



# Evaluation of idealized large-eddy simulations performed with the Weather Research and Forecasting model using turbulence measurements from a 250-m meteorological mast

Alfredo Peña[1], Branko Kosović[2], and Jeffrey D. Mirocha[3]

[1]DTU Wind Energy, Technical University of Denmark, Roskilde, Denmark
[2]National Center for Atmospheric Research, Boulder, USA
[3]Lawrence Livermore National Laboratory, Livermore, USA

**Correspondence:** Alfredo Peña (aldi@dtu.dk)

**Abstract.** We investigate the ability of the Weather Research and Forecasting model to perform large-eddy simulation of canonical flows. This is achieved through comparison of the simulation outputs with measurements from sonic anemometers on a 250-m meteorological mast located at Østerild, in northern Denmark. Østerild is on a flat and rough area, and for the predominant wind directions, the atmospheric flow can be considered to be close to homogeneous. The idealized simulated flows aim at representing atmospheric boundary layer turbulence under unstable, neutral, and stable stability conditions at the surface, which are statistically significant conditions observed at Østerild. We found that the resolved fields from the simulations appear to have the characteristics of the three stability regimes. Vertical profiles of observed mean wind speeds and direction are well reproduced by the simulations with the largest differences under near-neutral conditions, where the effect of the subgrid-scale model is evident on the vertical wind shear close to the surface. Vertical profiles of observed eddy fluxes are also well reproduced by the simulations with the largest differences for the three velocity component variances under stable stability conditions, although nearly always within the observed variability. With regards to turbulent kinetic energy, we find good agreement between observations and simulations at all vertical levels. Simulated and observed velocity spectra match very well, and show very similar behavior with height and with atmospheric stability within the low frequency interval; at the effective resolution, the simulated spectra show the typical drop-off of finite differences. Our findings demonstrate that these idealized simulations reproduce the characteristics of atmospheric stability regimes often observed at a high turbulent and flat site within a direction sector, where the air flows over nearly homogeneous land.

## 1 Introduction

For many applications and, in particular, for wind energy, we would like to characterize the long-term site conditions, i.e., first- and second-order statistics of the three-dimensional velocity vector, at a number of locations and vertical levels within a given area, so that we take into account all relevant motion scales of the atmosphere. For such a purpose, a multiscale modeling approach is needed, in which one starts by downscaling the large scales of atmospheric motions, from, e.g., reanalysis and global forecasts, to the regional or the mesoscales using a numerical weather prediction (NWP) model, and continuing down





to the microscales through forcing of (nesting to) a Reynolds-averaged Navier–Stokes-like or large-eddy simulation (LES)
domain.

Currently, there are a couple of atmospheric modeling systems capable to seamlessly simulate the spatiotemporal behavior
of the atmosphere at its multiple scales. One of such is the open-source community-open Weather Research and Forecasting
(WRF) model (Skamarock et al., 2008). The WRF model has for many years been used to dynamically downscale the large-
scale motions to the mesoscale and this capability has been highly exploited, i.a., for wind energy research (Peña and Hahmann,
2012; Hahmann et al., 2015; Kosović et al., 2020). The multiscale ability of the WRF model is mainly achieved by its grid
nesting capabilities and physical process parameterizations designed for different scales. In recent years with the increase in
computer power, attempts to further nest the WRF model down to the microscale, i.e., at spatial resolutions of $\approx 100$ m or less,
have been performed (Talbot et al., 2012; Rai et al., 2019). The high resolution domains can be run in the WRF model in a LES
mode (WRF-LES), i.e., large-scale turbulent stresses and fluxes are resolved and the effect of the filtered scales by the LES is
modeled via subgrid-scale models.

The value of a WRF-LES-based system can be evaluated by performing real-time multiscale simulations of the atmosphere
and subsequent comparison with historical observations (hindcasting). However, such evaluations can be misleading as many
aspects of the modeling system play a role in the outputs, such as the uncertainty of the forcing datasets, and for wind in
particular, the resolution and accuracy of the topographic inputs, e.g., the resemblance to reality of the land use characteristics
and the assignment of roughness length values to predefined land use categories. Due to the difficulty to discern modeling-
from system-related abilities when evaluating real-time simulations, it is important to evaluate results from atmospheric models
with observations from sites and conditions resembling canonical flows. If such flows can be observed, the WRF model can
be run in an idealized fashion so that the modeler has control on the initial atmospheric boundary layer (ABL) characteristics,
topographical inputs, and forcing (Moeng et al., 2007; Mirocha et al., 2018).

From analysis of sonic anemometer measurements distributed vertically on a 250-m meteorological tower at Østerild, in
Northern Denmark, Peña (2019) demonstrated that for a range of wind directions, long-term statistics on the observations
of winds and turbulence have close resemblance to those one expects for flow over flat and homogeneous conditions. The
measurements at Østerlid provide details on the turbulence structure of the atmosphere within the range of heights where
modern large wind turbines operate. Peña (2019) also found a clear dependence with atmospheric stability and height above
the ground of the mean wind, direction, and turbulence parameters. The objective of this work is to find out whether or not we
are able to reproduce the ABL characteristics at Østerild using WRF-LESs and, if positive, provide the community with a solid
foundation for the utilization of WRF-LES in historical multiscale ABL simulations.

In this study, we first present the methodology (Sect. 2) used for the analysis of WRF-LESs, which includes the selection
of flow cases from the Østerild dataset and the setup of the WRF model. Section 3 presents the results, where we first provide
details with regards to the statistics used from the simulated ABLs and later we show comparisons with observations of
simulated wind, direction, and turbulence parameters. We also show a comparison of turbulence spectra under the three selected
stability regimes. Finally, discussion and conclusions are given in the last section.



## 2 Methodology

We focus our analysis on the accuracy of resolved and modeled atmospheric flow parameters by WRF-LESs through comparison with observations. We include in the comparison vertical profiles of mean wind speed and direction, and velocity variances and covariances, as well as turbulence spectra at various vertical levels.

### 2.1 Selection of flow cases

The measurements at Østerild are described in detail in Peña (2019). For this study, we only use statistics based on 10-min periods of measurements performed with Metek USA-1 sonic anemometers deployed at 7, 37, 103, 175 and 241 m on the meteorological tower. Figure 1-left shows the distribution of atmospheric stability conditions close to the surface (using the sonic anemometer measurements at 37 m) within the close-to-homegenous sector at Østerild. We do not use the measurements at 7 m for this purpose as these are strongly influenced by the local topographical inhomogeneities (forest trees) near the mast (Peña, 2019).

We are interested in modeling three types of ABLs: near-neutral ($|z/L| \leq 0.05$) but referred to as neutral for simplicity hereafter, unstable ($-0.5 \leq z/L \leq -0.2$), and stable ($0.2 \leq z/L \leq 0.5$), where $z$ is the height and $L$ the Obukhov length (Obukhov, 1946). As illustrated, the surface stability conditions are predominately neutral, however, unstable and stable conditions are frequently observed. The observed ensemble-average surface heat flux was very close to zero ($-6.2 \times 10^{-5}$ K m s$^{-1}$), 0.0948 K m s$^{-1}$, and $-0.0276$ K m s$^{-1}$, under neutral, unstable and stable atmospheric conditions, respectively.

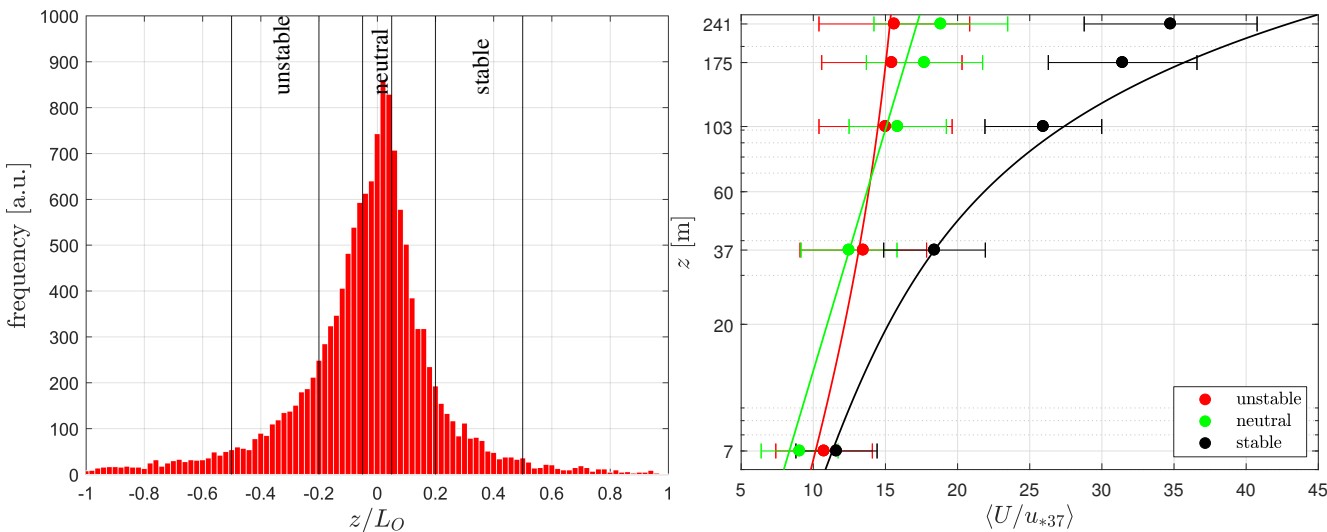

**Figure 1.** (Left) Frequency of atmospheric stability conditions observed at Østerild within the close-to-homogeneous sector. (Right) Normalized vertical wind profiles under different stability conditions. Observations are shown in markers and MOST profiles in solid lines (see details in the text). The error bars denote ±one standard deviation



Figure 1-right shows the behavior with height of the ensemble-average vertical profiles of the velocity magnitude $U = \sqrt{u^2 + v^2}$, where $u$ and $v$ are the along and crosswind components, respectively, normalized by the surface friction velocity $u_{*37}$, where 37 refers to the vertical level from the sonic-anemometer observations. The roughness length $z_0$ is estimated for each 10-min 'neutral' period as $z_0 = z \exp{(\kappa U / u_*)}^{-1}$, where $\kappa = 0.4$ is the von Kármán constant, using the observed values at 37 m. The ensemble average is estimated by taking the exponential of the mean of the logarithmic of $z_0$ 10-min values. This results in $z_0 = 0.2492$ m. For the unstable and stable conditions, the neutral value is too high and by using $z_0 = 0.0992$ m together with the atmospheric stability correction based on Monin-Obukhov similarity theory (MOST) (Monin and Obukhov, 1954), the prediction of the normalized vertical wind profile is very good for the first 100 m for the three main ABL regimes. We therefore choose these $z_0$-values for the LESs below. MOST predictions are given as

$$\frac{U}{u_*} = \frac{1}{\kappa}\left[\ln\left(\frac{z}{z_0}\right) - \psi_m\right], \tag{1}$$

and here we choose $\psi_m = 0$, $-4.7z/L$, and $-(3/2)\ln([1+(1-12z/L)^{1/3}+(1-12z/L)^{2/3}]/3)+\sqrt{3}\text{atan}([1+2(1-12z/L)^{1/3}]/\sqrt{3})-\pi/\sqrt{3}$ as in Gryning et al. (2007) for neutral, stable, and unstable conditions, respectively.

## 2.2 Simulations

We used the LES capability of the WRF model to perform idealized simulations. The WRF model is a non-hydrostatic, fully compressible solver of the Euler equations. It accomplishes LESs by turning off the planetary boundary-layer scheme options, and instead uses one of a number of subgrid-scale (SGS) models. Slip conditions for the horizontal velocity components are imposed at the model top, together with vanishing vertical velocity and fluxes.

## 2.3 Setup

Simulations were performed with the WRF model version 4.1.2 to simulate the ABL flow under neutral, unstable, and stable atmospheric conditions; thus we performed one simulation per ABL regime. We used a domain with $(n_x, n_y, n_z) = (500, 250, 120)$ grid points, where $x$, $y$, and $z$ are the horizontal and vertical directions, respectively. The model top was set to 2000 m. The horizontal resolution in both directions, $\Delta x$ and $\Delta y$, was 15 m. The vertical levels were chosen so that the grid aspect ratio, $\Delta x/\Delta z$, approaches a value of three close to the surface. The vertical spacing was kept constant up to about 250 m (covering the measurement levels of the mast), stretched out up to $\approx$900 m, where it reached 35 m, and kept constant upwards. Figure 2 illustrates the vertical grid spacing for each of the vertical levels. The idea of using the same domain setup for the three ABLs is to try to 'isolate' the ability of the model to simulate the particular atmospheric stability case.

The bottom of the domain is flat and the roughness length was set to the values estimated from the Østerild observations in Sect. 2.1 for each of the ABL types. The Coriolis parameter ($f_c$) was set to the value that corresponds to the mast location latitude (57.0489°). The time step used for the simulations was 0.1 s. All simulations were performed using the SGS model of Deardorff (1980) with the prognostic equation for the subgrid turbulent kinetic energy.

Simulations were initialized assuming a dry atmosphere. For the neutral ABL simulation, the initial temperature was kept constant (289.5 K) up to 700 m and then an inversion of 10 K km$^{-1}$ was imposed. Such an inversion strength is a common





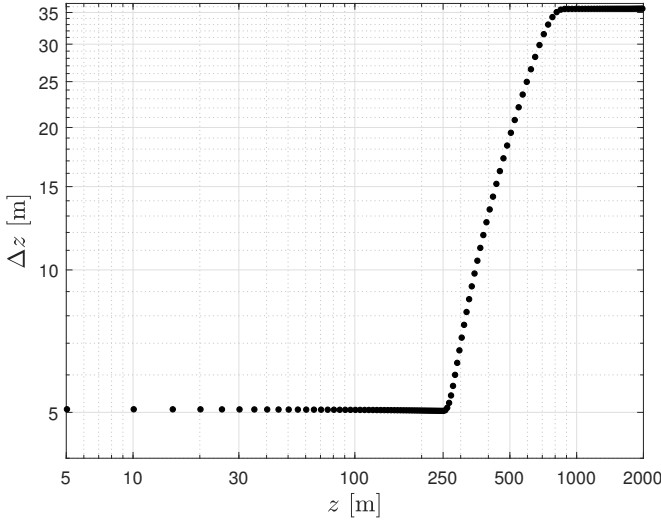

**Figure 2.** Vertical grid spacing as function of vertical level used in the simulations

choice for neutral ABL modelling (Pedersen et al., 2014; Mirocha et al., 2018). The height of the inversion was chosen to be lower (we expected the neutral ABL to slowly grow with time) than the value for the ABL-height estimation using the parametrization of Rossby and Montgomery (1935),

$$z_i = C \frac{u_*}{|f_c|}, \tag{2}$$

where $C = 0.1$–$0.5$. For a nearby site with similar climatology to Østerild, Peña et al. (2010a) found similar ABL heights by

comparing the estimations from Eq. (2) using $C = 0.15$ with those from observations of aerosol backscatter profiles under near-neutral conditions. Using the ensemble-average $u_*$-value from the near-neutral observations (0.69 m s$^{-1}$) and Eq. (2) with $C = 0.15$, $z_i = 845$ m. The initial $u_x$- and $u_y$-velocity components, aligned with the $x$- and $y$-axis, respectively, were kept constant throughout the ABL, with values of 14 and 0 m s$^{-1}$, respectively. For the unstable ABL simulation, the initial temperature was kept constant (289.5 K) up to 700 m (since we expected the unstable ABL to grow faster with time than the

neutral ABL) and then an inversion of 4 K km$^{-1}$ was imposed. Such an inversion strength is a common choice for unstable ABL modelling (Pedersen et al., 2013; Mirocha et al., 2018). The initial $u_x$- and $u_y$-velocity components were kept constant throughout the ABL, with values of 8 and 0 m s$^{-1}$, respectively. For the stable ABL simulation, the initial temperature was kept constant (289.5 K) up to 100 m and then an inversion of 10 K km$^{-1}$ was imposed. Such an inversion strength and level are common choices for stable ABL modelling (Kosović and Curry, 2000; Muñoz-Esparza and Kosović, 2018). Using

the ensemble-average $u_*$-value from the stable observations (0.36 m s$^{-1}$) and Eq. (2) with $C = 0.12$ as suggested for stable conditions in Peña et al. (2010a), $z_i = 353$ m. The initial $u_x$- and $u_y$-velocity components were kept constant throughout the ABL, with values of 14 and 0 m s$^{-1}$, respectively. The initial $u_x$-velocity for the three simulations was chosen so that it was close but slightly higher than the observed ensemble average of $U$ at each of the stability regimes at the highest vertical level.



MOST was applied at the surface through the in-built WRF surface-layer scheme (option 1 in WRF's namelist), although
a modification of the open-release scheme was performed to maintain simulations dry. The neutral and unstable ABLs were
simulated during 20 and 6 h by imposing a constant surface heat flux $\overline{w'\Theta'}$ of 0 and $0.0948 \, \text{K m s}^{-1}$, respectively, mimicking
the ensemble-average observed values. For the stable ABL, imposing a surface heat flux is problematic because it does not
guarantee a stable solution for the computed $u_*$-values (Basu et al., 2008). Therefore, imposing a cooling rate boundary
condition at the surface is a choice (Kosović and Curry, 2000). We apply a rate of $-0.25 \, \text{K h}^{-1}$, run for 24 h, and check the
ability of the model to reach the observed heat flux and friction velocity at the surface.

The simulations were performed using periodic boundary conditions in both horizontal directions. We output selected variables for a vertical column in the middle of the domain every 1 s and instantaneous values of those variables every 1 h for the
positions in the whole domain.

## 3 Results

For the three ABL regimes under study, we present an analysis of the simulated transient outputs (Sect. 3.1), an overall picture
of the simulated turbulence structures (Sect. 3.2), intercomparisons between simulated and observed vertical profiles of the
wind speed and direction (Sect. 3.3) as well as turbulent fluxes and kinetic energy (Sect. 3.4), and intercomparisons between
simulated and observed velocity spectra at different vertical levels (Sect. 3.5). For the intercomparison of vertical profiles, we
quantitatively assess the skill of the simulations by computing the root mean square error (RMSE) between simulations and
observations across the observed heights. RMSEs are computed by linearly interpolating the simulations to the five vertical
levels with sonic anemometer observations. Similarly, RMSEs are computed between the MOST profiles (Eq. 1) and the
observations at the five vertical levels.

### 3.1 Statistics on transient simulation outputs

We need to extract WRF-LES outputs to perform the comparison with the observed statistics at Østerild. The choice of the time
to extract LES statistics depends on the type of boundary layer. The analysis is mostly made by performing moving averages
over 600-s windows based on the 1-Hz outputs of given variables. We estimate the height of the ABL $z_i$ as that in which the
maximum of the vertical gradient of potential temperature occurs. $u_*$ is an output of the WRF model, which is computed within
WRF's surface-layer scheme,

#### 3.1.1 Neutral conditions

Figure 3-left illustrates the time series of $u_*$, $z_i$, and SGS turbulent kinetic energy $e_{\text{sgs}}$ on the first model level (indicated by
the subscript). It is seen that turbulence was triggered slightly before 2 h. $u_*$ behaves similarly to $e_{\text{sgs},1}$; the former directly
depends on the resolved velocity closest to the ground. Although with nearly steady moving averages, the estimated $z_i$ slightly
increases during the simulation after $\approx$5 h.





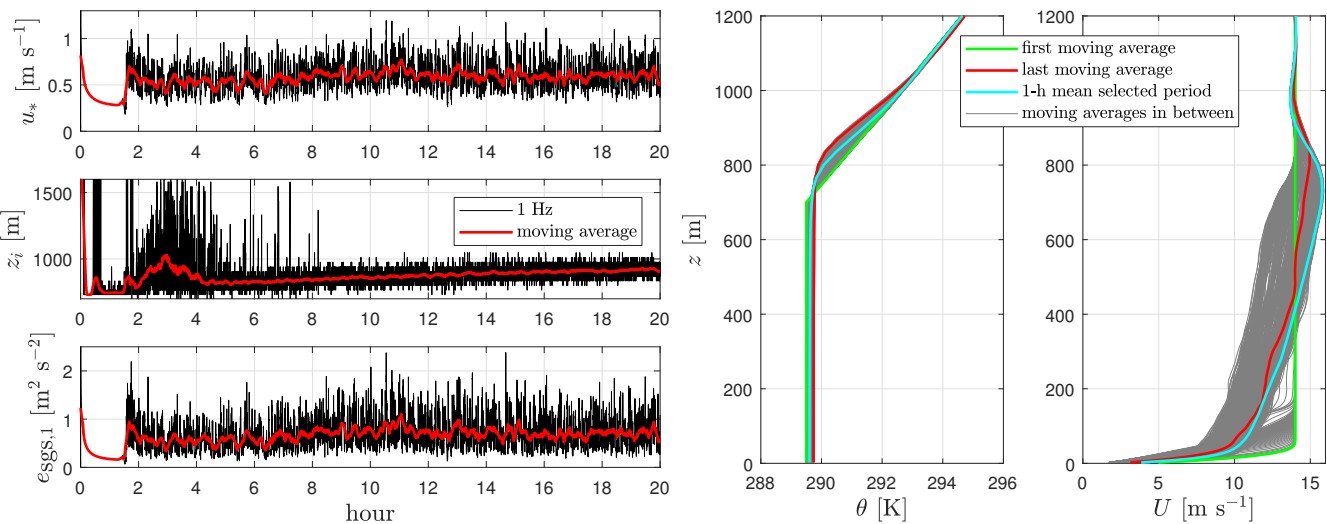

**Figure 3.** (Left) Time series of selected WRF-LES outputs and estimated variables for the neutral ABL simulation. (Right) A number of moving averages of vertical profiles of potential temperature and horizontal wind speed magnitude during the simulation. The mean vertical profiles for the selected 1-h period are also highlighted

Figure 3-right shows a number of moving averages during the 20-h simulation of vertical profiles of potential temperature $\Theta$

155 and horizontal wind speed magnitude $U = \left(u_x^2 + u_y^2\right)^{1/2}$. It is observed that the height of the potential temperature inversion slightly increases, and so does $z_i$. It is also observed that the wind speed becomes supergeostrophic during the simulation with a maximum between 10 and 11 h. We therefore chose this hour for computing the neutral ABL WRF-LES statistics.

### 3.1.2 Unstable conditions

Figure 4 illustrates the same information as Fig. 3 but for the unstable ABL simulation with the correspondent imposed

160 positive heat flux at the surface. In Fig. 4-left, it is seen that turbulence was triggered much earlier compared to the neutral ABL simulation. $u_*$ is generally lower than the values of the neutral ABL simulation mostly because of the lower $z_0$ imposed. $z_i$ increases faster with time compared to the neutral ABL simulation and is above 1000 m for the latest 3 h. $e_{\mathrm{sgs},1}$ is lower than that of the neutral ABL simulation because the geostrophic forcing is significantly lower than in the neutral case.

In Fig. 4-right, it is clear the effect of the positive surface heat flux on the temperature profile, which explains the behavior

165 of $z_i$. The wind speed is, as expected, much more constant with height above $\approx$100 m compared to that of the neutral ABL. To compute the unstable ABL WRF-LES statistics, we select the output within 3–4 h because $z_i$ is very close to 1000 m, and so it is slightly higher than that of the neutral ABL simulation, and the mean wind speed is very constant (slightly below 7 m s$^{-1}$) and subgeostrophic up to about 1100 m, as expected due to the stable layer at the top, and increases about 1 m s$^{-1}$ within the next 150 m reaching the geostrophic value.





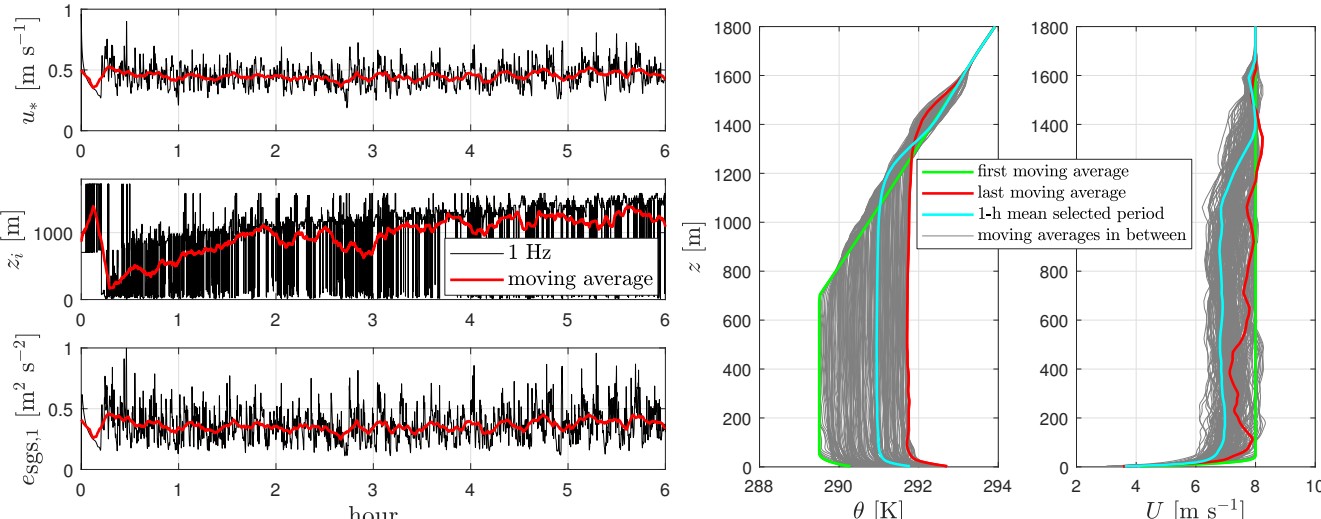

**Figure 4.** Similar to Fig. 3 but for the unstable ABL simulation

### 3.1.3 Stable conditions

Figure 5 illustrates the same information as Figs. 3 and 4 but for the stable ABL simulation, which was performed by imposing a surface cooling rate. We also illustrate time series of the first model level horizontal speed $U_1$ and surface heat flux to further show the closeness of the latter to the observed surface heat flux at Østerild. In Fig-5-left, it is seen that turbulence was triggered slightly before 2 h similar to the neutral ABL simulation. After $\approx$8 h, the moving average of $u_*$ becomes relatively steady and slightly higher than the observed value at 37 m. After turbulence is triggered, $z_i$ slightly increases until its moving average reaches a value $\approx$300 m at 8 h and remains nearly steady afterwards. $U_1$, $u_*$, and $e_{\mathrm{sgs},1}$ reach nearly stationary state after approximately 8 h. About the same time, the ABL reaches stationary state as evident by the values of $z_i$. The moving average of $\overline{w'\Theta'}$ decreases nearly linearly until it reaches the observed value at 37 m a little before 8 h and remains rather steady and generally slightly higher than the observed value thereafter. Note that we verified in the time series output that the surface potential temperature decreased at the imposed cooling rate of $-0.25$ K h$^{-1}$ (not shown).

In Fig. 5-right, we clearly see the effect of the cooling rate on the temperature profile. The initial imposed inversion quickly 'disappears', a strong inversion develops with time within the range $\approx$275–400 m, and the initial inversion strength of 10 K m$^{-1}$ is recovered thereafter. As for the neutral ABL simulation, the wind speed becomes supergeostrophic with the nose at the height where the strong inversion starts. We select the range 15–16 h to compute the stable ABL statistics.

### 3.2 Instantaneous resolved fields

Figure 6 shows instantaneous cross-sections of $U$ along the $x$-$z$ plane at the $y$-direction midpoint and along the $x$-$y$ plane at $z \approx 100$ m for the three types of ABL. Similar to Mirocha et al. (2018), we find elongated low-speed structures along the



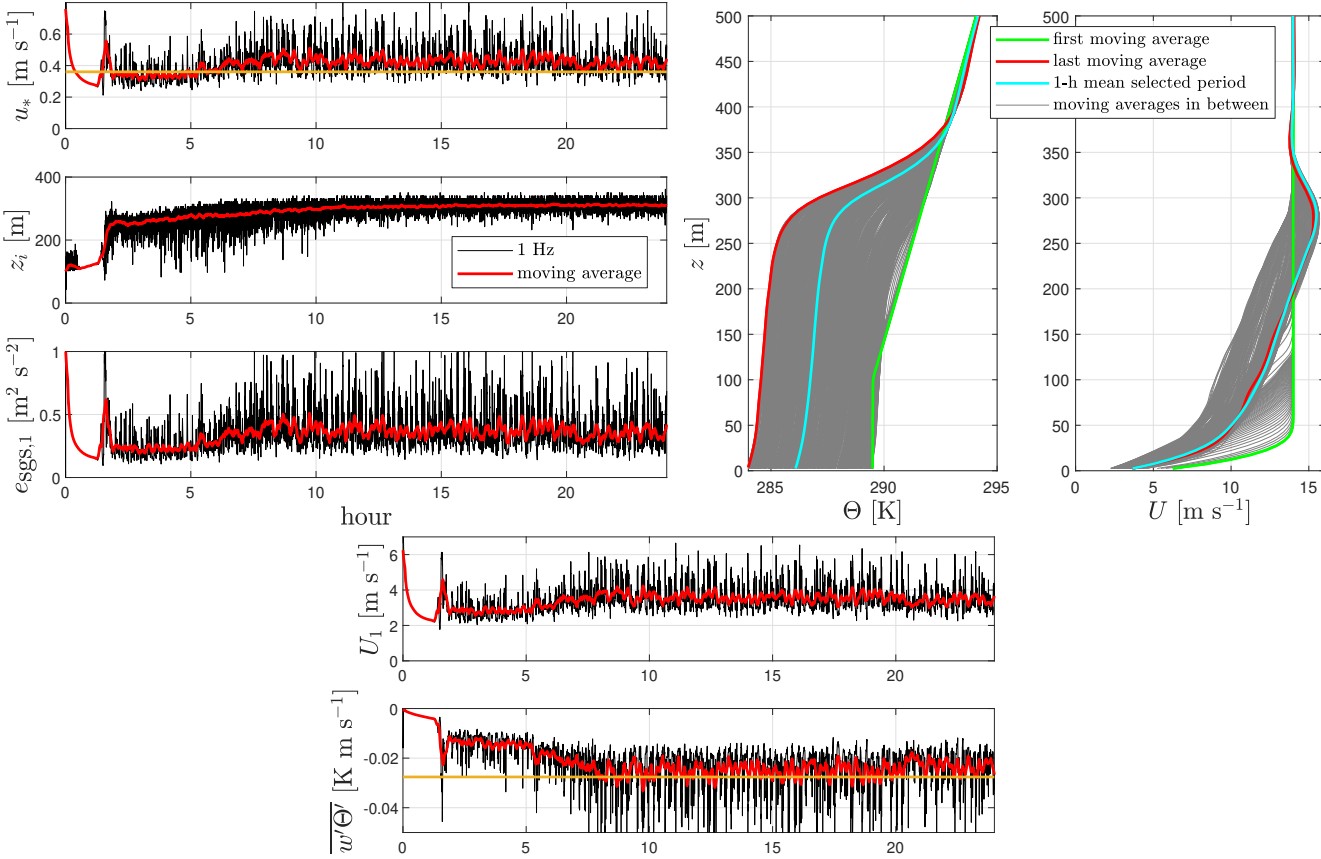

**Figure 5.** Top frames similar to Fig. 3 but for the stable simulation. (Bottom) Time series of other selected WRF-LES outputs for the stable simulation. The solid line in ochre shows the observed friction velocity and observed heat flux

streamwise direction and turbulence structures of different sizes all up to the capping inversion for the neutral simulation. For the unstable simulation, the turbulence structures are less elongated along the streamwise direction compared to the neutral ABL simulation and wave-like structures appear beyond the local inversion. For the stable simulation, the elongated turbulence structures along the streamwise direction are more pronounced and narrower compared to the other ABL types and are well confined below the capping inversion.

### 3.3 Wind speed and direction profiles

Figure 7 shows the behavior of the normalized wind speed $U/u_*$ within the first 1200 m and within the measurement range (in a semilogarithmic plot) for the three types of ABL. In general, qualitatively, the simulations show good agreement with the observations, particularly for the stable ABL because the simulated jet is just above the highest observed level and so the high vertical wind shear from the observations is well captured.

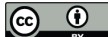

**Figure 6.** Instantaneous cross-sections ($x$-$z$ through the middle of the domain and $x$-$y$ at $\approx$100-m height) of $U$ at 13 h for the neutral (left), at 3 h for the unstable (right), and at 15 h for the stable ABL simulation (bottom). Note that the $x$-$z$ cross-section for the stable ABL simulation has an aspect ratio of 1:4 and $z$ is limited to 500 m



**Figure 7.** Normalized simulated and observed wind profiles for neutral (top two most left), unstable (top two most right) and stable conditions (bottom). MOST profiles (Eqn. 1) are also shown. The error bars and dashed lines denote ±one standard deviation from the mean of the observations and simulations, respectively



All simulations match well the observed value closest to the surface, i.e. that at 7 m. When looking within the bulk of the measurement range, we see the strongest deviations from the simulations compared to the mean of the observations under the neutral ABL case. Particularly, within the first ≈40 m from the ground, the neutral simulation overpredicts the normalized wind shear due to the tendency of the specific SGS model to overpredict the dimensionless shear (Mirocha et al., 2018). For the three cases, the mean of the simulations is always within the observed variability, which is larger than that of the simulations.

RMSEs of the normalized simulated wind speeds also reflect the qualitative findings: 1.800, 0.821, and 1.467 for neutral, unstable, and stable conditions, respectively. RMSEs of the normalized MOST profiles (1.059, 0.445, and 4.157 for neutral, unstable, and stable conditions, respectively) are lower for the neutral and unstable ABLs, and much higher for the stable ABL compared to those from the simulations. For the stable ABL, MOST already overpredicts the wind at 103 m, as expected due to the shallow surface layer, which can roughly be estimated as 10% of the ABL height, i.e. ≈35 m. The comparison between the simulations and MOST is however not fair. MOST was used in Sect. 2.1 to estimate the surface roughness length under neutral conditions; thus MOST profiles should match fairly well the observed normalized wind speed within the surface layer (and perfectly at 37 m). The simulations do not know a priori the observed normalized wind speed at any vertical level and use the surface roughness length value as a boundary condition only.

Figure 8 shows the behavior of the turning of the wind within the measurement range in a semilogarithmic plot for the three types of ABL. The observations show the largest turning of the wind under stable conditions and the lowest under unstable conditions, as expected. The neutral simulation is the one that differs the most from the observations (RMSE of $2.49°$) as within the measurement range the simulated wind does not turn much; however, most of the turning occurs higher up (not shown) and at the top of the ABL, the relative turning is $23°$, while this is $15°$ and $30°$ for the unstable and stable simulation, respectively. RMSEs are lowest under unstable conditions ($0.52°$) and under stable the RMSE is rather low ($1.47°$). As for the wind speed, the mean of the simulations for the three cases is always within the observed variability. The simulated variability is clearly highest under unstable conditions.

## 3.4 Turbulent fluxes and kinetic energy

For the comparison with the measurements, we need to estimate the total variances and covariances from the WRF-LESs; thus we need to account for the resolved and the subgrid stresses. The total stress is given as

$$\tau_{ij}^{\text{tot}} = \tau_{ij}^{\text{res}} + \tau_{ij}^{\text{sgs}}, \tag{3}$$

where $\tau_{ij}^{\text{res}}$ is the resolved stress, i.e., $-\rho\overline{u_i'u_j'}$, and $\tau_{ij}^{\text{sgs}}$ the subgrid stress. The latter can be computed as,

$$\tau_{ij}^{\text{sgs}} = (2/3)\delta_{ij}e_{\text{sgs}} + \tau_{ij}^{\text{dev}}, \tag{4}$$

where $\tau_{ij}^{\text{dev}}$ is the deviatoric part of the subgrid stress, which is an output of the SGS scheme in the WRF model. Note that the resolved turbulent kinetic energy $e_{\text{res}}$ is $\overline{u_i'u_i'}/2$ (with implicit summation) and so the total turbulent kinetic energy is the sum of both SGS and resolved terms. Since the observed statistics are computed so that $u$ is aligned with the wind direction at each vertical level for each 10-min period, we need to rotate both the simulated velocities and stresses to align them with the simulated direction for each simulated vertical level.





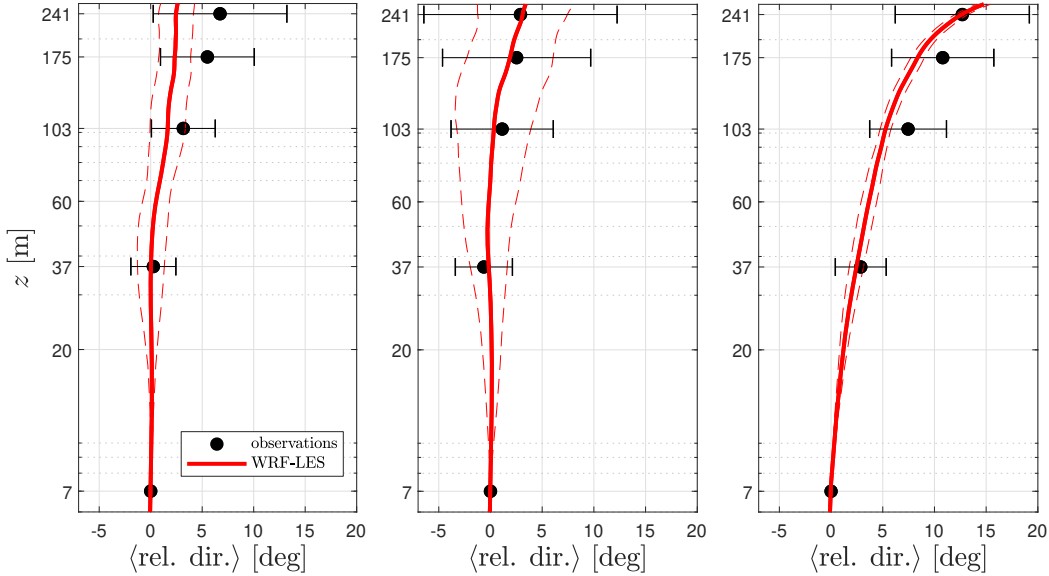

**Figure 8.** Observed and simulated turning of the wind for neutral (left), unstable (middle), and stable (right) conditions. The error bars and dashed lines denote ±one standard deviation from the mean of the observations and simulations, respectively

Figure 9 shows the vertical profile of the simulated and observed eddy fluxes within the measurement range for neutral conditions. Observations and simulations of the velocity variances are in relatively good agreement, particularly at $\geq 100$ m, and the largest apparent differences are found for the $uw$-covariance, where the resolved value is higher than that of the observations at $\geq 100$ m and is lower than the observed value at $\leq 100$ m. For both the $uw$- and $vw$-covariances, the SGS term
seems to overcompensate the flux when compared to the first observed level. For the $u$-variance, the resolved term is already close to the observed value and accounts for most of the total variance, whereas close to the surface the SGS term strongly aids both $v$- and $w$-variances in matching the observed fluxes.

For unstable conditions (see Fig. 10), the apparent bias between observations and simulations is in line with that for neutral conditions for both variances and covariances. For the variances, the apparent bias is slightly higher at $\geq 100$ m and lower
at lower levels, whereas it is generally higher for the $uw$-covariance and lower for the $vw$-covariance when compared to the results under neutral conditions. Note that the observed variability of fluxes is also higher under unstable compared to neutral conditions.

For stable conditions (see Fig. 11), the apparent bias between observations and simulations is generally the highest among all stability conditions for the three velocity variances, and it is low at the two observed levels closest to the ground. Note
that the observed normalized $u$- and $v$-variances do not decrease much with height compared to their behavior under neutral and unstable conditions but all simulated normalized velocity variances show a faster decrease with height compared to the simulations under neutral and unstable conditions. The apparent bias for the $uw$- and $vw$-covariances is comparable to that found under neutral and unstable conditions. For the three stability conditions, in general, the observed variability is larger than





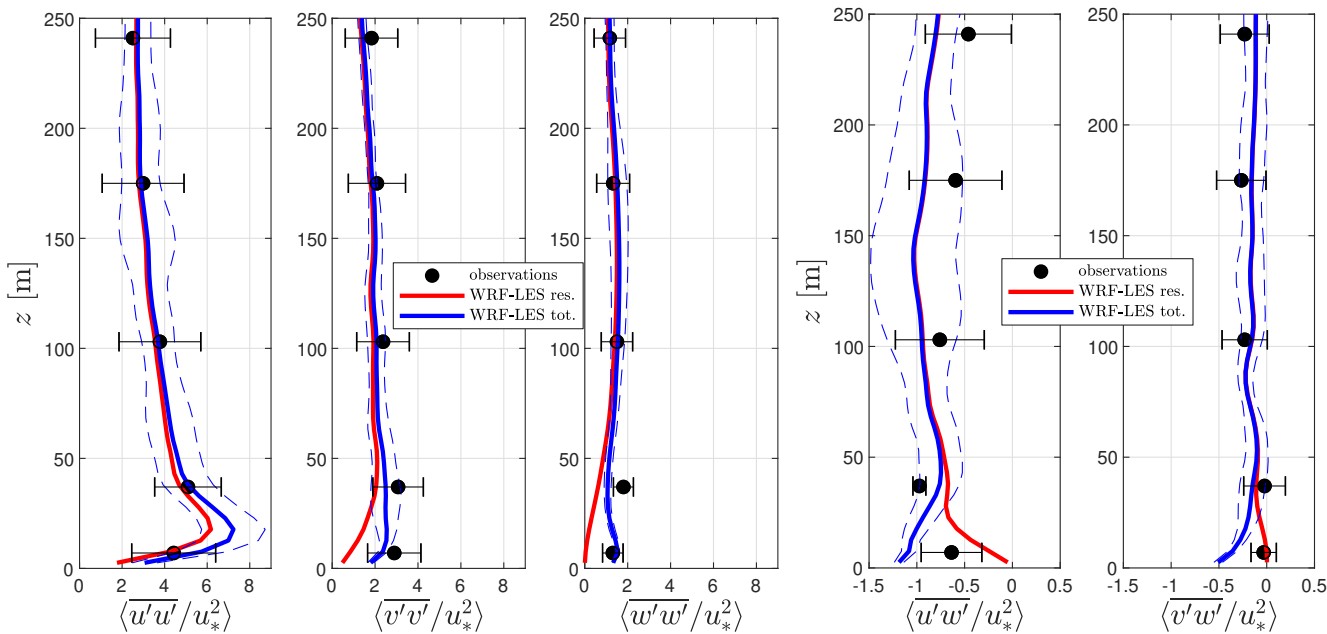

**Figure 9.** Simulated and observed vertical profiles of normalized velocity variances (three most left) and covariances (two most right) under neutral conditions. Resolved and total fluxes are shown for the simulations. The error bars and dashed lines denote ±one standard deviation from the mean of the observations and simulations, respectively

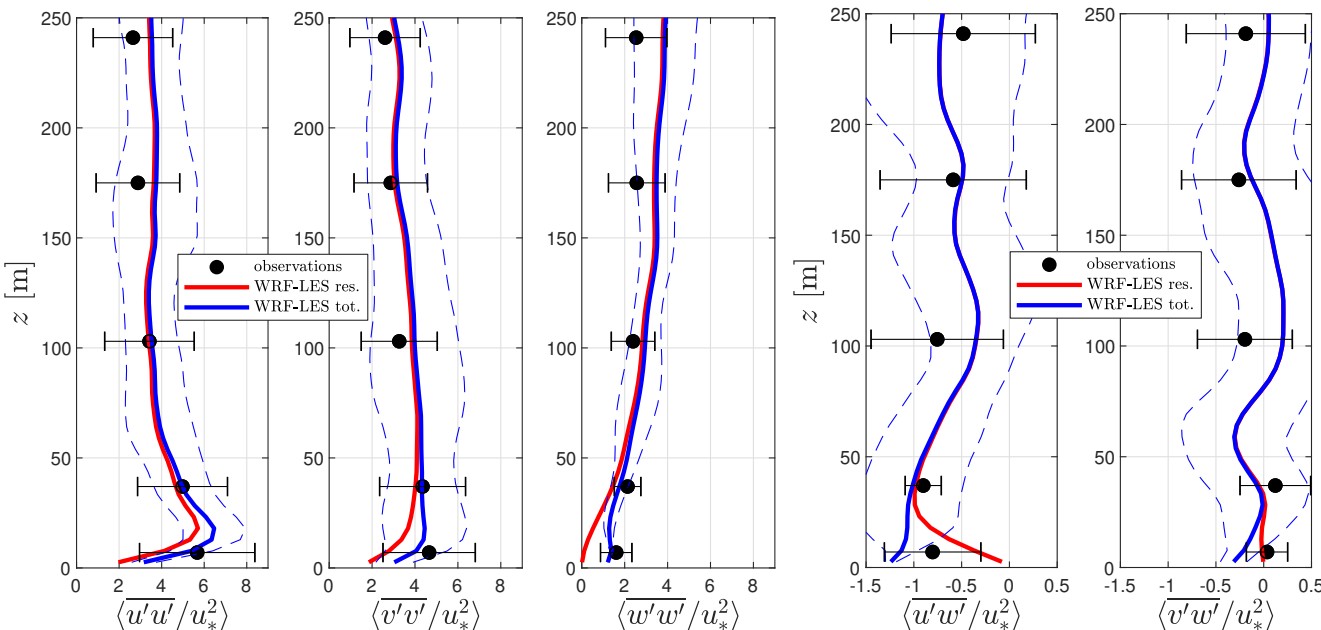

**Figure 10.** Similar to Fig. 9 but for unstable conditions





that of the simulations when looking at the three velocity variances; however for neutral and unstable conditions, they are close
to each other when looking at the $uw$- and $vw$-covariances.

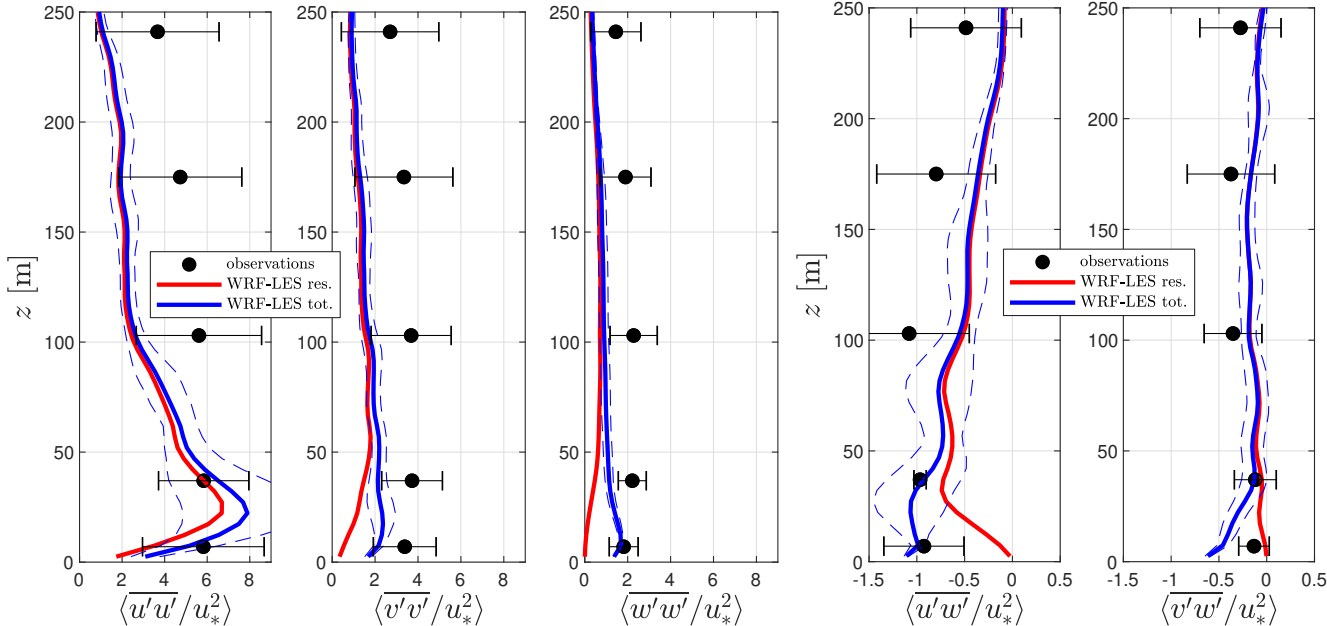

**Figure 11.** Similar to Fig. 9 but for stable conditions

Table 1 shows the RMSEs of the normalized simulated turbulent fluxes for the three ABL regimes, where stable conditions
present the largest values (except for the normalized $vw$-covariance). Note that for all the three stability regimes, the RMSEs of
the normalized velocity variances are higher than those of the normalized velocity covariances because the velocity variances
are larger than the velocity covariances. Despite of this, under neutral conditions the RMSE for the normalized $u$-variance
(0.144) is lower than that of the normalized $uw$-covariance (0.275) when accounting for the observed heights other than 7-m
only. For unstable and stable conditions, this also occurs for specific vertical levels.

**Table 1.** RMSEs of the normalized simulated turbulent fluxes for each of the ABL types

| ABL type | $\langle\overline{u'u'}/u_*^2\rangle$ | $\langle\overline{v'v'}/u_*^2\rangle$ | $\langle\overline{w'w'}/u_*^2\rangle$ | $\langle\overline{u'w'}/u_*^2\rangle$ | $\langle\overline{v'w'}/u_*^2\rangle$ |
|---|---|---|---|---|---|
| neutral | 0.463 | 0.443 | 0.353 | 0.320 | 0.181 |
| unstable | 0.616 | 0.552 | 0.797 | 0.268 | 0.260 |
| stable | 2.209 | 1.750 | 1.054 | 0.352 | 0.214 |

Figure 12-left shows the ratio of the SGS to the total turbulent kinetic energy as function of height for the three types
of ABLs. As expected, the percentage of the SGS term in the total is highest for stable conditions and lowest for unstable
conditions. Further, more than 10% of the energy comes from the SGS term below 120 m under stable conditions, which is





about 40% of the ABL height. For neutral and unstable conditions, the SGS term contributes more than 10% of the total energy within 10% and 3% of the ABL height, respectively.

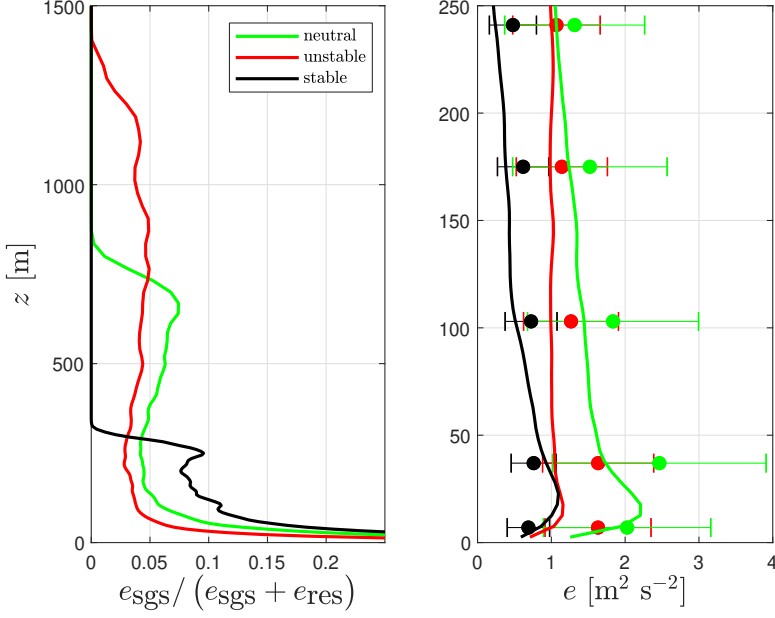

**Figure 12.** (Left) vertical profiles of the ratio of the subgrid to the total (subgrid plus resolved terms) turbulent kinetic energy from the simulations of the three types of ABLs. (Right) simulated (solid lines) and observed (markers) vertical profiles of total turbulent kinetic energy. The error bars denote ±one standard deviation from the mean of the observations

Figure 12-right shows the vertical profile of total turbulent kinetic energy under the three ABL regimes and for both simulations and observations. Within the measurement range, the highest simulated and observed values are found under neutral conditions, since this is the regime in which we observed the highest roughness length value that is used as bottom surface
condition in the simulations. It is noticed that the results of the three simulations are within the observed variability and that it is under stable conditions, where the bias on the mean value is the lowest. RMSEs are also lowest for stable ($0.211$ m$^2$ s$^{-2}$) when compared to neutral ($0.412$ m$^2$ s$^{-2}$), and unstable ($0.415$ m$^2$ s$^{-2}$) conditions.

### 3.5 Turbulence spectra

Figure 13 shows power spectra of the three velocity components at four vertical levels (the different frames) under neutral
conditions for both the observations and the simulations. The simulated spectra is computed from the simulated output at the vertical level closest to the sonic anemometer. The observed spectra is the ensemble average of the 10-min power spectra for all the 10-min periods in which neutral conditions are observed. The simulated spectra are computed by diving the 1-Hz output over the selected hour into 51 10-min periods (overlapping over 540 s). The 51 power spectra are then ensemble averaged.

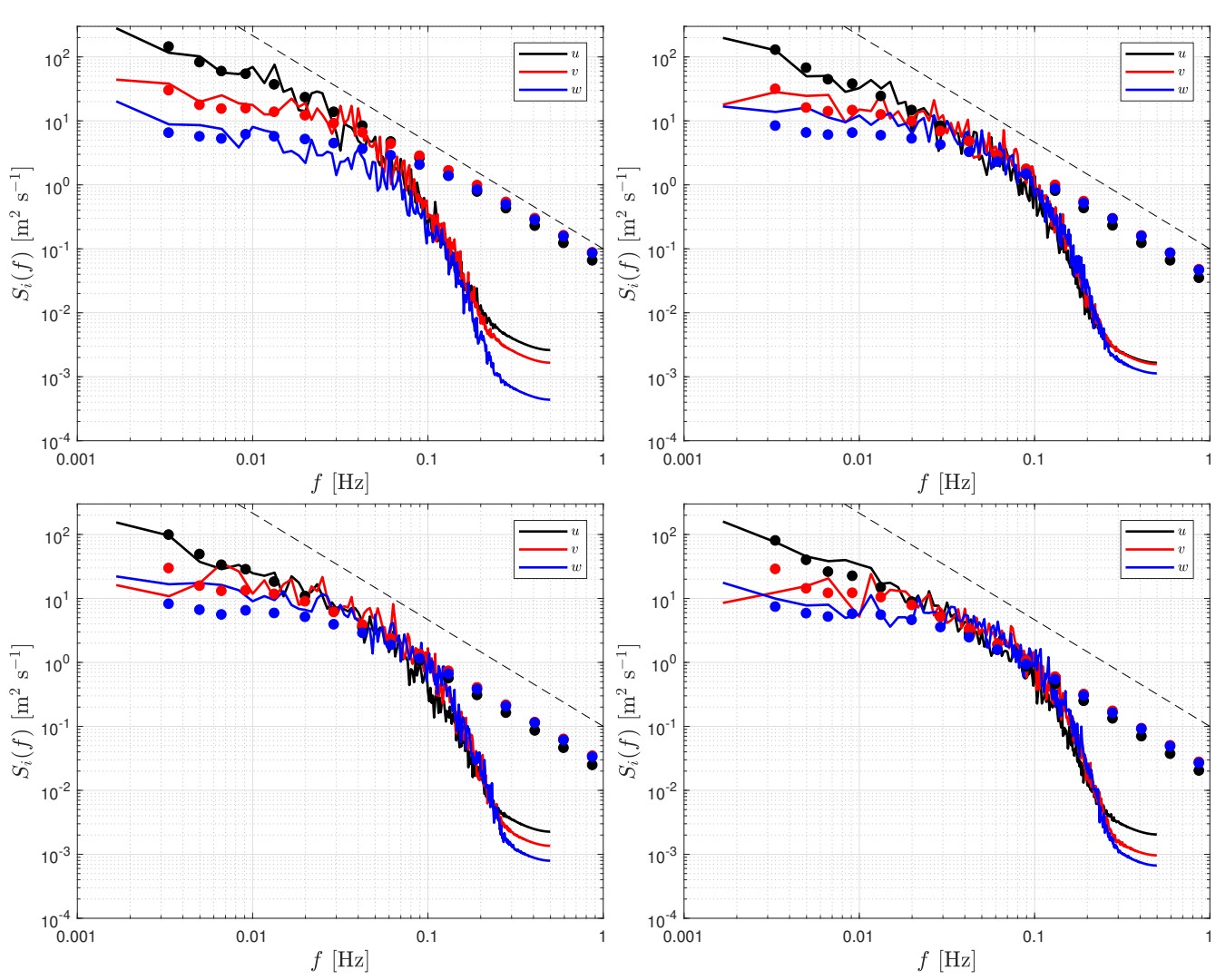

**Figure 13.** Neutral velocity spectra at 37 (top left), 103 (top right), 175 (bottom left), and 241 m (bottom right). Markers indicate observations and the solid lines simulations. The dashed line represents a slope of $-5/3$





As shown, the observed power spectra is very well behaved at all vertical levels with an inertial subrange slope close to $-5/3$ following Kolmogorov (1941). From the lowest frequencies up to $\approx 0.1$ Hz, both simulated and observed spectra are very close, which explains the good agreement between simulated and observed velocity variances in Sect. 3.4. From $\approx 0.1$ Hz, a drop-off of the velocity spectra appears, which is typical of finite difference and discretization schemes (Skamarock, 2004). The effective resolution is $\approx 7\Delta x = 105$ m, which corresponds to a frequency of $\approx 0.1$ Hz within the wind speed range 9–12 m s$^{-1}$. It can also be observed that at 37 m, the drop-off occurs at a frequency lower than 0.1 Hz and this drop-off frequency slightly increases with height, as expected.

Under unstable stability conditions (Fig. 14), the observed velocity spectra also approaches the inertial subrange slope of $-5/3$. Compared to the results for neutral conditions, it is also clearer both when looking at the simulated and the observed spectra that turbulence becomes more isotropic the higher the vertical level and the more unstable the atmosphere is, as the three velocity spectra become close to each other.

Under stable stability conditions (Fig. 15), we also find the $-5/3$ slope on the observed velocity spectra. When looking both at the simulated and observed spectra, we can see a clear distinction between the velocity spectra compared to unstable conditions as turbulence is more anisotropic; by fitting the spectral three-dimensional turbulence model of Mann (1994) to the observed velocity spectra and $uw$-cospectra from the sonic anemometer measurements at Østerild, Peña (2019) found a distinct lower turbulence anisotropy the more unstable the surface conditions were, whereas neutral conditions appeared to be slightly more anisotropic than stable conditions. Within the low frequency range and under stable conditions, we can notice more flattened spectra compared to that under unstable and neutral conditions. The premultiplied spectra, i.e., $fS_i(f)$ (not shown), peaks at higher frequencies under stable compared to unstable conditions, which translates into turbulence length scales that are larger under unstable compared to stable conditions (Peña et al., 2010b).

# 4    Conclusions and discussion

To assess the ability of high-fidelity simulations, such as LES, to reproduce the behavior of winds and turbulence within the first hundreds of meters of the ABL, which is useful, e.g., for the siting of wind turbines, we need to try, first, to isolate the effects of physics parametrizations and forcing, and, second, to analyze high-quality measurements of both wind and turbulence at several heights. The reason for the former is that such parametrizations and forcing conditions influence the behavior of turbulence and so it is difficult to differentiate their effects, which are accounted for, e.g., in real-time simulations using mesoscale models, from those inherent to the abilities of LESs. Here, by using wind and turbulence statistics, and velocity spectra computed from sonic-anemometer measurements on a 250-m mast over the predominant wind direction at Østerild, Denmark, we demonstrate that idealized WRF-LESs reproduce well the observed wind and turbulence characteristics, which resemble canonical flow of typical ABL regimes (unstable, neutral, and stable).

Comparison with observations reveals that under the three ABL regimes, the vertical profiles of normalized wind and direction are well reproduced by the simulations. The simulated means are always within the observed variability. Within the first $\approx 40$ m from the ground, the mean vertical wind shear of the neutral simulation is much higher than the observed mean,

**Figure 14.** Similar to Fig. 13 but for unstable conditions

**Figure 15.** Similar to Fig. 9 but for stable conditions



which has also been found in previous studies when SGS models of the same type are used. Within the measurement range, the simulated wind turns the lowest under neutral conditions, whereas the observations show the highest turning under stable and the lowest under unstable conditions, as expected. However, the simulated wind turning within the depth of the ABL under neutral conditions is between those of the two other ABL regimes.

Vertical profiles of observed normalized eddy fluxes are also well reproduced by the simulations. For nearly all vertical levels and for the three ABL regimes, the simulated values are within the observed variability. Under neutral conditions, in particular, the simulated mean normalized velocity variances have an excellent agreement with the observed means specially above 50 m, the best agreement is found under unstable conditions below 100 m, and under stable conditions, there is a systematic underestimation of the mean observed values by the simulations above 50 m. For the normalized $uw$- and $vw$-covariances, the agreement between simulations and observations is generally better for the latter, and for unstable conditions.

Vertical profiles of observed turbulent kinetic energy reveal the highest values under neutral conditions, as expected, due to the high roughness value that was estimated from the observations using MOST, and lowest under stable conditions. The simulations show the same behavior, although the mean values for both unstable and stable conditions are much closer to each other compared to the observed values. This is because we use the same boundary condition (roughness length) for the unstable and stable simulations, and so the surface-layer scheme in the WRF model computes similar $u_*$ values for both regimes. The observations, on the other hand, reveal a much higher value for $u_*$ at the 37-m height under unstable compared to stable conditions. The simulations show systematically lower values than the observations, although within their variability.

Simulated and observed velocity spectra match very well within frequencies lower than that correspondent to the effective resolution, which explains the good agreement between simulated and observed velocity variances. Such a good match is found both under the three ABL regimes and the vertical levels examined. As expected due to the nature of the WRF model, the velocity spectra shows a drop-off close to the effective resolution and so it is only the observed spectra the one that approaches the $-5/3$ slope within the inertial subrange. Both simulated and observed velocity spectra show that turbulence is more anisotropic the more stable the ABL and the closer to the surface; the more sheared the flow, the more anisotropic the turbulence.

Regarding the assumptions made for the simulations we carried out, it is appropriate to reiterate that these are idealized simulations. As such, the initial conditions may not represent observed cases. Observations of the ABL height, and observations of vertical profiles of both potential temperature and water vapor mixing ratio within the extent of the ABL are not available at Østerild. The three cases considered here are all characterized by relatively weak surface heat fluxes and strong shear, i.e., they are shear driven. Therefore, assuming a dry atmosphere, i.e., that the moisture effects on the structure of the atmospheric surface layer are small, is a good approximation in the three cases. Since these are idealized simulations, the initial potential profile is well mixed up to 700 m for the neutral and convective boundary layers, and 100 m for the stably stratified boundary layer. Capping inversions develop naturally due to surface heating or cooling, while the overlaying inversion in the free troposphere is specified. The overlying inversion ($10\ \mathrm{K\ km^{-1}}$), which we used for the neutral and stable ABLs, corresponds to the standard adiabatic lapse rate for dry atmosphere. These choices are commonly used in idealized simulations as well as a value lower



than 5 K km$^{-1}$ for the unstable ABL. In all three cases, the ABL height evolves during the simulation based on the combined effect of shear and buoyancy forcing.

Given that the comparisons between the outputs of the idealized simulations and the observed statistics are rather good, in future studies we want to explore the ability of a WRF-LES-based multiscale modeling system to simulate in real-time the

ABL at Østerild and other sites in which high-quality measurements are also available, in more typical unsteady operating conditions. Key issues to address for such purposes include the smoothing effect on turbulence when forcing LES domains with mesoscale information, the modeling of turbulence in intermediate domains when nesting down from mesoscale to LES, and the inherent difficulties of the WRF model to simulate atmospheric flow over terrain steeper than 30–40 deg.

Significant progress has been made already in the development of methods to accelerate the development of three-dimensional

turbulence in LES domains nested within mesoscale simulations, in neutral (Muñoz-Esparza et al., 2015) and non-neutral (Muñoz-Esparza et al., 2017; Muñoz-Esparza and Kosović, 2018) boundary-layer settings, as well as in wind energy applications, one featuring a mesoscale frontal passage interacting with a portion of an operating wind plant (Arthur et al., 2020b). Additional validation of these approaches using a similar framework to that applied herein will further establish WRF's value to wind energy applications.

Another important element of multiscale atmospheric simulation involves downscaling through the "gray zone" or "terra incognita" scales (e.g., Wyngaard, 2004). New approaches, based on scale-awareness (e.g., Honnert et al., 2011), explicit three-dimensional turbulence transport (Kosović et al., 2020), and explicit filtering and reconstruction (Simon et al., 2019) represent promising pathways, but also require further evaluation in wind energy applications.

The WRF model's applicability over steep terrain, a known issue when downscaling due to topographic features being better

resolved, is likewise being extended, using both higher-order numerical methods (Arthur et al., 2020a) as well as immersed boundary methods (Lundquist et al., 2012; Arthur et al., 2018). These methods have likewise not been adequately evaluated in relation to wind energy relevant flow information.

The framework of the present study can be used to assess the utility of the WRF model in these above described settings to improve wind energy simulations in a broader range of real-world environments and operating conditions, for which

smaller-scale flow information, including turbulence, in relation to environmental and meteorological variability, is invaluable to supporting the continued expansion of the wind energy industry.

*Code and data availability.* The numerical outputs were generated with the open-source WRF model (https://github.com/wrf-model/WRF). Both the observational and simulated data intercompared in the manuscript, as well as the input files for the WRF model simulations are available at: https://figshare.com/s/c02c0954051d67f17992 (Peña, 2020).



*Author contributions.* AP performed the simulations, analyzed both the simulation outputs and observational data, and drafted the manuscript. All authors were involved in the design of the numerical experiments and the proposed methodology. All authors contributed to the revision and finalization of the paper.

*Competing interests.* The authors declare that they have no conflict of interest.

*Acknowledgements.* This work is partly funded by the Ministry of Foreign Affairs of Denmark and administered by Danida Fellowship
Centre through the 'Multi-scale and model-chain Evaluation of Wind Atlases' (MEWA) project. JDM's contribution is supported by LLNL under Contract DE-AC52-07NA27344 and by the U.S. Department of Energy's Wind Energy Technologies Office.



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
