# Peer review of "Evaluation of idealized large-eddy simulations performed with the Weather Research and Forecasting model using turbulence measurements from a 250-m meteorological mast"

_Wind Energy Science, 2020_

## Author Comment (AC1)

.

**Response to the comments from referee 1**

Thanks for your comments on our manuscript. Here our response to each of your comments. The response is given in blue color.

Best regards,
The authors
* * *
**General comment**

Very complete assessment of the WRF-LES model in idealized conditions covering mean profiles and spectra at a 250-m boundary-layer mast. The main challenge in the study is in the inherent inconsistencies in the comparison of canonical ABL simulations with ensemble-averaged profiles. The study is based on the assumption that the observed profiles represent the horizontally homogeneous conditions that the LES model is based upon. This is not convincingly addressed or quantified so it deserves some additional discussion. The validation is quantified using normalized quantities based on u*. A suggestion is made to use other set of normalized profiles to avoid the intrinsic bias in u* in the computation of the RMSE.

Thanks for your general comment. We address how representative our measurements are when compared to canonical flows when responding to your first specific comment and your suggestion on normalizing the profiles in a different manner later when responding to the related specific comment.

**Specific comments**

P3. Section 2.1: I think you should justify why doing long-term statistics leads to mean profiles that can be considered good references for idealized ABL simulations. While bin averaging with long-term data will definitively help in the convergence of the mean profiles you don't guarantee that you are removing persistent heterogeneities from the mesoscale wind climate. To come up with reference profiles for canonical ABL you would want to filter out situations that deviate from quasi-steady and horizontally homogeneous conditions. Have the authors considering any filtering for such conditions? By filtering more you can decrease the value of the std-bars in the observed profiles to values that are closer to the ones from LES (e.g. Fig. 7 shows this discrepancy in the error bars). Even if you provide a reference about the measurements, please describe the case selection in more detail with regards to the binning process and number of samples in each case and discuss how you deal with these issues.

As the reviewer points out, we cannot guarantee that there are no mesoscale trends in the observations and as the reviewer suggests, one way of filtering out those periods is by deriving mesoscale tendencies from real-time WRF simulations. However, this is beyond the scope of the paper as we only performed idealized simulations. A discussion of this possibility is now added in the Discussion section. Following the recommendations of the reviewer, we now also extend Section 2.1 (with the description of the selection of the flow cases from the measurements) so that information with regards to the binning process, extension of the analyzed periods, and the amount of cases per stability condition is provided. It is important to note that in the error bars we show the standard deviation and not the standard error, which due to the large number of samples is very small. This is because here we are interested in the variability of both observations and simulations. Note also that the measurements extend over a 4 year period and so the statistics on each stability regime are robust. Filtering can indeed increase the variability of the observations; in Fig. 1, we show that when we only use 10-min periods that are between three consecutive 10-min periods showing the same flow characteristics (a simple stationary filter), the variability highly increases for neutral conditions. For these conditions, about a third of the data are filtered out with such a filter, and for stable and unstable conditions is nearly a fourth and one-eighth of the data, respectively. Also, if there was a strong mesoscale trend in the observations, it will be reflected in the low frequency range of the velocity spectra. However, it is particularly in that range where both simulations and observations compared the best. This is also now part of the discussion in the revised manuscript.

P4,83: Can you discuss if these stability functions have been found at the Østerild site and if they are also used in the surface layer model of WRF? Together with the roughness length, this may be relevant in the interpretation of surface fluxes.

These functions have been tested at Østerild (actually as it can also be seen in the normalized wind profiles in

[Figure]

Figure 1: Normalized wind profile at Østerild with a simple non-stationary filter

Fig. 1 in the manuscript). And these functions were also used in the work of Peña (2019) at Østerild. This is now stated in the manuscript. As for WRF, the surface-layer scheme uses the same function under unstable conditions but a slightly stronger stability correction under stable conditions. However, at the first model level (≈5 m), the difference between the correction from WRF and from this function is lower than 2%.

P4.Setup: There is no mentioning of any sensitivity analysis on the grid settings that would support the selected grid configuration. While the setup looks reasonable it would be better to include some discussion on the adequacy of these settings for canonical ABL simulations. In particular, it would be interesting to compare with the parametric study of LES model settings done by Mirocha et al (2018) that also includes idealized WRF simulations.

We did not perform grid sensitivity studies in a systematic matter and used this grid resolution and configuration based on the grid sensitivity study by Mirocha et al. (2018). They also found that the optimal grid aspect ratio $\Delta x/\Delta y = 3$ provided good comparison with two other models and with observations. As suggested, we now state this in the manuscript.

P5.Figure 2: The x-axis is the vertical level height. For those not familiar with WRF grid, please indicate if this height represents the cell center or the cell height.

As suggested, we now add this information in the caption of the figure.

P5.113: Can you discuss how you come up with the values of geostrophic wind (14 m/s in neutral and stable conditions and 8 m/s in unstable conditions)? I would suggest using a table to collect all the input quantities that you use to define the three flow cases.

This information is provided in Sect. 2.3 Setup: "The initial $u_x$-velocity for the three simulations was chosen so that it was close but slightly higher than the observed ensemble average of $U$ at each of the stability regimes at the highest vertical level.". As suggested, we now add a table collecting the inputs for each of three flow cases.

P6.144: "The choice of the time to extract LES statistics depends on the type of boundary layer." Can you elaborate on how you select this time? Based on the discussion later on, the selection seems a bit arbitrary although I understand that the profiles never reach a steady state. Maybe, at this point, you should discuss this challenge. My impression is that you end up choosing profiles that have sufficiently developed to a quasi-steady state close to the values you have from measurements and "have the right look" for canonical ABLs.

As suggested, at the end of Sect. 3.1.3 we now discuss the challenge on the selection of the time due to the inherent type of simulation: "It is important to note that it is challenging to select the time for extracting the statistics due to the inherent unsteadiness of LES. For neutral conditions, we can objectively choose the period, but for unstable and stable conditions, this is a combination between reaching a quasi-steady state and the fluxes values observed at the surface mainly."

P7.Fig 3: Can you indicate (in the caption or in the legend) which time in the simulation corresponds to the blue line? A vertical line in the time series would also help.

*As suggested, we now indicate in vertical lines the time of the selected period on the time series plots in Figs. 3–5*

Figs 3/4/5: The TKE behaves similarly to the u* so you might as well skip it. In neutral conditions you may consider replacing it with the wind speed of the jet nose since this is used to select the neutral profile. In unstable and stable conditions you may consider replacing u* by the heat flux (as in Fig 5), or the Obukhov length (I'd use this one), to see how the profile evolves from the energy point of view.

*As suggested, we now replace the tke by the jet nose speed time series in the neutral case. We also replace the tke by the heat flux time series in the unstable case. For the stable case, we now skip the tke and first model wind speed time series as well. The text has been modified accordingly.*

P8.172: You should mention that the first model level is at around 5 m while the reference flux measurements are taken at 37 m. This difference may be significant in the value of the fluxes, specially in stable conditions. Why not using the closest level to 37 m (in all the plots) as the reference height for the time series? Why not showing the observed values of u* and heat flux in Figs. 3 and 4? In principle, instead of using the u* value produced by WRF at the surface, you could compute u* from the high-frequency time series as if they were sonic measurements to try to represent the same quantity (as it is done in section 3.4 for the fluxes). To improve consistency, you could introduce filters in the measurements to remove mesoscale trends and mimic the behavior of the sgs model in LES (as the low-pass filter displayed in the spectra). I understand that this is beyond the scope of the paper at this point but I think it is worth discussing how "canonical" are the measurements when we compare with idealized LES simulations since these quantities integrate different turbulent scales in simulations and measurements.

*As suggested, we now remind the reader (at the end of the first paragraph in Sect. 3.1) that the observed fluxes are at 37-m height, whereas the surface simulated fluxes are derived from the first model level (about 5 m). We choose to show these simulated fluxes (and not those close to 37 m) because of what we have now further elaborated in Sect. 2.1 Selection of flow cases: by using the the 37-m fluxes at Østerild, the behavior of the dimensionless wind shear with dimensionless stability follows surface-layer scaling. Also, as suggested, we now show the observed fluxes in Figs. 3–5. It is important to notice that the roughness values are estimated based on the observed quantities at 37-m height as surface-layer scaling holds there. In the LES, surface-layer scaling is applied at the first model level and so although the heights between observations and simulations are not the same, the method to derive roughness (from observed fluxes) and that to simulate fluxes (from roughness) is consistent. This is also added in the text.*

P11.Fig. 7: Because of the differences in the reference heights of $u_*$ (and the underlying filter in LES), you are probably introducing a bias in the normalized profiles. Although less formal, maybe a simpler but more robust way of normalizing the profiles would be using the wind speed at 37 m instead of the friction velocity.

*Normalizing by 37 m does not reduce the error bars much at the levels most far apart (237 m) but the idea of scaling with $u_*$ is also to test that there is consistency between the observed and simulated fluxes (see the response to the previous comment).*

P2.203: Please put the metrics of this section in a Table or put them together with those in Table 1.

*As suggested, we now add the metrics in Table 1.*

Figures 9,10,11; Table 1: As discussed before, these plots and error metrics depend on quantities that are normalized with the u* value, which may come with a bias due to using a different vertical level in measurements and observations. In addition, as pointed out by the authors, u* in the simulations depend on the value of the roughness length that is used as input which is difficult to quantify in reality. For these reasons, I think it is better to choose a reference height and normalize each profile by the corresponding value of each quantity at that reference height, e.g. $U/U_{37}$, $uw/uw_{37}$, etc (and exclude the 37 m level in the computation of RMSE).

*Please, see the response to the the last three comments.*

P18, 305: "The simulated means are always within the observed variability"... well, the variability is pretty large in the measurements, isn't it? Still, the agreement is pretty good considering the uncertainties in the

definition of "canonical" flow cases.

As suggested by the reviewer we now replace the sentence by: "The simulated means are always within the observed variability, but it should be noted that the latter is large; the observed variability at Østerild is lower than that from the 'canonical' flow observations performed at the 200-m tower at the SWiFT test facility in the US Southern Great Plains (Mirocha et al., 2018)"

P22.344: When running real-time simulations you can definitively extract mesoscale tendencies from WRF that you can use to filter out periods of strong heterogeneity. This can help you narrow down the case selection to obtain mean profiles that more closely match the idealized conditions of this study.

This is a good idea, which we will consider for future studies. This is also now discussed in the Discussion section

**References**

Mirocha, J. D., Churchfield, M. J., Muñoz Esparza, D., Rai, R. K., Feng, Y., Kosović, B., Haupt, S. E., Brown, B., Ennis, B. L., Draxl, C., Sanz Rodrigo, J., Shaw, W. J., Berg, L. K., Moriarty, P. J., Linn, R. R., Kotamarthi, V. R., Balakrishnan, R., Cline, J. W., Robinson, M. C., and Ananthan, S.: Large-eddy simulation sensitivities to variations of configuration and forcing parameters in canonical boundary-layer flows for wind energy applications, Wind Energ. Sci, 3, 589–613, 2018.

Peña, A.: Østerild: a natural laboratory for atmospheric turbulence, J. Renew. Sustain. Energ., 11, 063 302, 2019.

---

## Author Comment (AC2)

.

**Response to the comments from referee 2**

Thank you very much for the comments on our manuscript. Here our response to each of your comments. The response is given in blue color.

Best regards,
The authors
* * *
**General comment**

The paper presents a validation of WRF LES model on three canonical flow cases: neutral, stable, and unstable. The motivation, structure, content, graphical presentation, and the referenes, are all excellent.

The high-quality data for the validation is collected at Østerild test centre, from 5 heights between 7 and 240 metres.

The three flow cases were defined using the Monin-Obukhov stability at 37 meters above the ground. The surfaces heat fluxes and/or surface temperature tendency, and the roughness length, resulting in the observed variables, were used to set-up the WRF model. This approach is adequate, but the assumption of the homogeneous surface should be validated to ensure that the flow characteristics are not too much affected by the fetch distance for different heights and wind speeds. It is possible that the apparent roughness length would be different if other heights than 37 metres would be used to derive the necessary parameters.

As the reviewer points out, the roughness length is different when computed from different heights. This was illustrated in Peña (2019). In the same study, it was also shown that the behavior of the dimensionless wind shear, $\phi_m = (\kappa z/u_*)(dU/dz)$, with stability, $z/L$, follows closely surface-layer scaling over a homogeneous, flat surface when looking at the 37-m height instead of 7-m. We now extend Sect. 2.1 (where we first describe the measurements at Østerild) with this information

Given that a lot of effort has been invested into estimations of the surface roughness using satellite and lidar data, especially around Østerild, one could ask why you haven't used the available roughness data for these WRF simulations. Are you suggesting that the roughness length is a function of the flow, and not the other way around?

The surface roughness length is a function of the flow, at least in the light of the logarithmic law of the wall (it is actually an integration constant). This is not very helpful for modeling as we need some inputs and boundary conditions to bound the flow, the roughness length being one of them. As we want the simulations to represent the observed conditions, we need to extract the roughness information from the observed flow behavior. This is the approach used here. But, looking at the bigger picture, the idea is that satellite/lidar-derived roughness maps do also reflect roughnesses consistent with those one can extract from the flow observations so that we can use those products for flow modeling elsewhere. In this work, however, our focus is not on the goodness or accuracy of those products.

The model's capability to accurately simulate the three flow cases is nicely analyzed in terms of the mean properties, and all stress components. The agreement between the LES model and real data is quite good. In particular the velocity spectra are fantastic, in the resolved part of the spectrum of course.

All three cases are shear-driven which, especially for the stable case, somewhat limits the applicability of the results. You have proposed a few directions for further work and we can look forward to it being performed and presented.

Thank you very much for your general comment about our analysis. We will hopefully provide the community with further contributions on this topic soon.

**Specific comments**

1. What was the period of the data collection, i.e. how much data is actually used in the aggregation for U, N, and S? From Figure 1 it seems that about 6 months of data is used, however from L274 it follows that there is

3000 seconds of data. Please clarify.

Following the recommendations of the reviewer, we now also extend Section 2.1 (with the description of the selection of the flow cases from the measurements) so that information with regards to the binning process, extension of the analyzed periods, and the amount of cases per stability condition is provided.

2. Figures 3-5: would it be possible to estimate the Ri number for these cases? It could be useful for comparisons when more of the flow cases will be constructed in the time to come.

The issue with the Ri number is that it is very sensitive to the levels used for its calculation. As shown in the new version of the manuscript, both simulated fluxes (in the forms of friction velocity and surface heat flux) approach rather well the observed values, and so the cases can be readily classified based on the surface fluxes, i.e., on the dimensionless stability $z/L$ (Fig. 1-left).

3. Figure 6: it would be easier to compare the instantaneous flow representations if the color scales were the same.

As suggested by the reviewer we tried that but setting the color scales within the same range either distorts the flow structures or disappears some of them, as the wind speeds are not the same for the three conditions

4. P12L203: Here the RMSE is introduced, but it is not immediately clear if this is the statistics derived from 5 values (5 heights, and using the whole-period-average values), or? Please clarify.

As suggested by the reviewer, at the end of the first paragraph in Sect. 3 (where we first presented RMSEs), we add a sentence clarifying that the RMSEs are based on the mean of the values of the examined quantity. In the previous sentences in that paragraph, it is stated that RMSEs are computed between simulations and observations across the observed heights. We now also add 'five' between 'the observed' to further clarify this.

5. P12L214: Please elaborate/comment on the possible reasons why the neutral simulation differs most from the observations, i.e. the stable and unstable simulations match the observations better.

As suggested, we now extend the discussion within that paragraph: "The overprediction under neutral conditions of the simulated dimensionless wind shear within the first tens of meters from the surface is the result of an overprediction of the simulated vertical shear of the $u$-component. Thus, the contribution of the $v$-component to the turning of the wind diminishes, which results in low values for the relative direction."

6. Figures 9-11 (also elsewhere but perhaps easiest to discuss here): the differences between N and S cases are very small, it requires quite an effort to find any significant differences. Have you considered finding "more stable" flow cases, or would that be difficult at this location?

More stable cases are much less frequent at Østerild as observed from the frequency of stability conditions in Fig. 1-left. We do not fully agree with the reviewer in that the neutral and stable cases are similar: as already noticed from the observations of normalized winds in Fig. 1-right, the wind shear in stable conditions is much higher than that in neutral conditions. The observed friction velocity under neutral conditions is nearly doubled that of stable conditions and the neutral TKEs are more than twice the values in stable conditions.

7. P16L262-267, Figure 12: Please expand the discussion of the TKE profiles. For example, the neutral simulation seems to "compress" the TKE profile towards the surface, evident by the "nose" of the profile being quite lower than 37 metres, which is the height of the maximum observed value

As suggested by the reviewer, we now expand the discussion in that paragraph: "In the three ABL regimes, there is a 'local maximum' in the total turbulent kinetic energy profile close to the surface. This shows the limitation of the LES to resolve turbulence below the local maximum; from this level down to the surface, the SGS contribution increases substantially."

8. P16L272: diving $->$ dividing.

Changed as suggested.

9. P21L339-340: there is some confusion regarding the inversion strength, and the adiabatic lapse rate for dry atmosphere. The capping inversion and the adiabatic lapse rate should have the opposite signs, and are

unrelated in this context. What was the purpose of the statement in L339-340?

As suggested, we now delete the comment on the adiabatic lapse rate as it was confusing.

**References**

Peña, A.: Østerild: a natural laboratory for atmospheric turbulence, J. Renew. Sustain. Energ., 11, 063 302, 2019.